# Quality Tests of Hybrid Joint–Clinching and Adhesive—Case Study

Jakub Kowalczyk [1,*], Waldemar Matysiak [2], Wojciech Sawczuk [1], Daniel Wieczorek [1], Kamil Sędłak [3] and Michał Nowak [2]

1   Faculty of Transport and Civil Engineering, Institute of Machines and Motor Vehicles, Poznan University of Technology, 60-965 Poznan, Poland
2   Faculty of Mechanical Engineering, Institute of Material Technology, Poznan University of Technology, 60-965 Poznan, Poland
3   Faculty of Mechanical Engineering, Division of Virtual Engineering, Poznan University of Technology, 60-965 Poznan, Poland
*   Correspondence: jakub.kowalczyk@put.poznan.pl; Tel.: +48-61-665-2248

**Abstract:** Inseparable joints are widely used in machine and vehicle construction. Hybrid joints include bonding with sheet metal clinching. This combination reduces costs as well as the time of production compared to welded joints. Tests on the samples made of DC01 sheets were carried out. A case study was conducted on four research series. For each series, the shear forces of the joint were measured. The first series consisted of adhesive bonding, and the second and third series consisted of hybrid bonding, during which the sheet metal clinching joint was developed immediately after the completion of adhesive application and after full joint formation. The last test series only includes sheet metal clinching. In the series where bonding was used, the homogeneity of the prepared joints was analysed using the ultrasonic echo technique. The shear strength of the bonded joints was 476 N, whereas the shear strength of sheet metal clinching was 965 N. For the hybrid joint, the average forces were 1085 N (for the specimens in which the lap joint was made after the joint was fully cured) and 1486 N (for the specimens in which the lap joints were made immediately after the adhesive was applied). It was discovered that the clinching of the steel sheets significantly increases the strength of the joint. The stabilisation of the joint causes better crosslinking conditions. This results in an increase in the strength of the hybrid joint.

**Keywords:** adhesion; ultrasound; strength; connection quality; sheet metal clinching

## 1. Introduction

Adhesive bonds are widely used in automotive manufacturing. The scope of their application is wide and includes the assembly of windscreens and rear windows, connecting hinges to the car body, fixing brake pads to load-bearing plates, and the connecting elements of the car body: door, roof, engine cover, and the boot. Bonding not only makes it possible to achieve a permanent connection, but it also seals the structure, dampens vibrations, and reduces vehicle production costs. Bonding allows us to join different materials, such as glass and metal [1]. The main limitations of adhesive bonding are the relatively long setting time for structural adhesives and lower mechanical strength than in the case of welded joints. To overcome the above disadvantages, joining methods such as welding, clinching, and riveting can be used in conjunction with adhesive bonding. Welded joints in the aspect of manufacturing technology and quality control are relatively well-known [2,3]. Similarly, the technology of sheet metal clinching is frequently applied [4–6]. On the other hand, adhesive joints are used just as often today. Several electromagnetic techniques have been introduced and developed (and are currently constantly improved) that enable non-destructive testing of their quality, such as infrared thermography, microwave, and terahertz imaging [7–11]. Nevertheless, the methods of quality control of adhesive joints

using ultrasonic testing [12–14] and fabrication [14] are well-known and commonly used in technical applications. The use of adhesive bonds results not only in reduced production costs but also in reduced vehicle mass, resulting in reduced combustion and exhaust gas emission into the atmosphere. Joining technologies in motor vehicles are being improved all the time. These works include such connections as welding [15–17], bonding [12–18], and welding and clinching [19–23]. Sheet metal clinching is carried out by a compression method, using small values of lateral forces on the die side and no adverse deformations near the joint [24–27]. The joint may be oval in shape and consist of cup-shaped indentations on the die and punch side. The joint is purely plastic in nature without compromising the cohesion of the material. This method makes it possible to permanently join parts mainly of thin sheets and to achieve high static and dynamic joint strength. Sheet metal clinching consists of cold pressing the sheets to be joined with a round punch and a suitably shaped die. The result of this process is a connection in the form of a round point, which is not subject to further processing and is free of burrs and sharp edges (Figure 1). The deformation does not affect the surface of the materials to be joined. The technology of joining by compression allows joining metal sheets in such a way that in the area of their overlap—by means of plastic deformation (displacement, elongation, swelling)—fasteners are formed without the thermal loading of connected parts. The sheets involved in the joining can be of different thicknesses and materials. Spatial fasteners are formed in the area of the die and punch action of a given forming tool. The depth of clinching determines the nominal convexity of the fastener on the die side and the elongation of the cup wall on the punch side. The bottom thickness characterises the degree of swell as well as the strengthening of the material in the joint area. It is a dimension of the part and a parameter describing the joining process. The most widespread use of this type of sheet metal joining has found its application in electrotechnics and radiotechnics, as well as in the construction of instruments in precision mechanics.

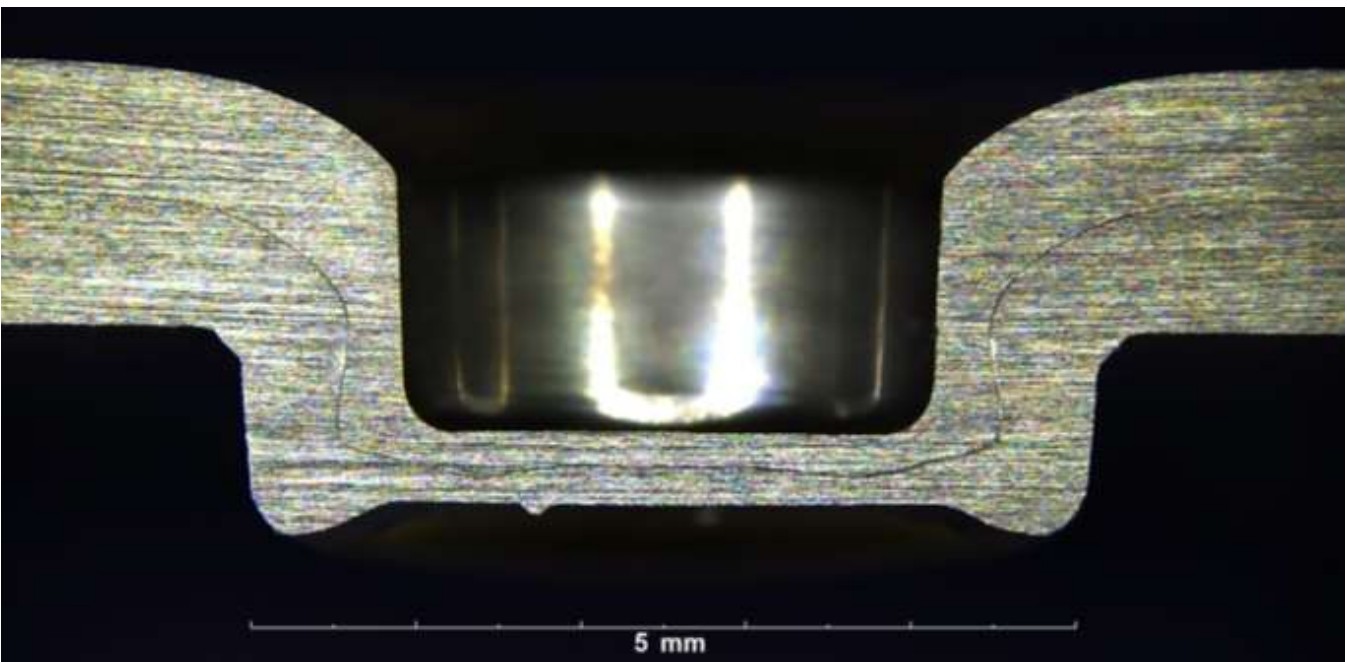

**Figure 1.** Cross-section of the sheet metal clinching.

The corrosion protection surface of the sheet metal is not affected. A slight bulge of material on the die side, which is acceptable in most applications, can be eliminated, if necessary, by using a different die shape. The saving in joining costs compared with spot welding is 30–60%, and the plates do not require any complicated preparatory processes. This method can be used to join sheets of different materials (carbon steel, stainless steel,



aluminium, etc.) and different thicknesses (up to 9 mm); coated, painted (e.g., powder coated), galvanised, enamelled, covered with a film layer, felt, or paper. The connection shows very high resistance to static and dynamic loads, and depending on the degree of automation, it takes only a few seconds.

An area that has not been fully recognised is hybrid connections type metal clinching and bonding. Hybrid joints are joints that use different joining techniques [28]. Such joints are not only used in vehicle manufacturing but are also increasingly utilised in the area of sheet metal repair. For a long time, the classic riveting method has been used in car body repairs. The disadvantage of this solution is the necessity of drilling a hole and the possibility of making connections only in non-visible areas (Figure 2).

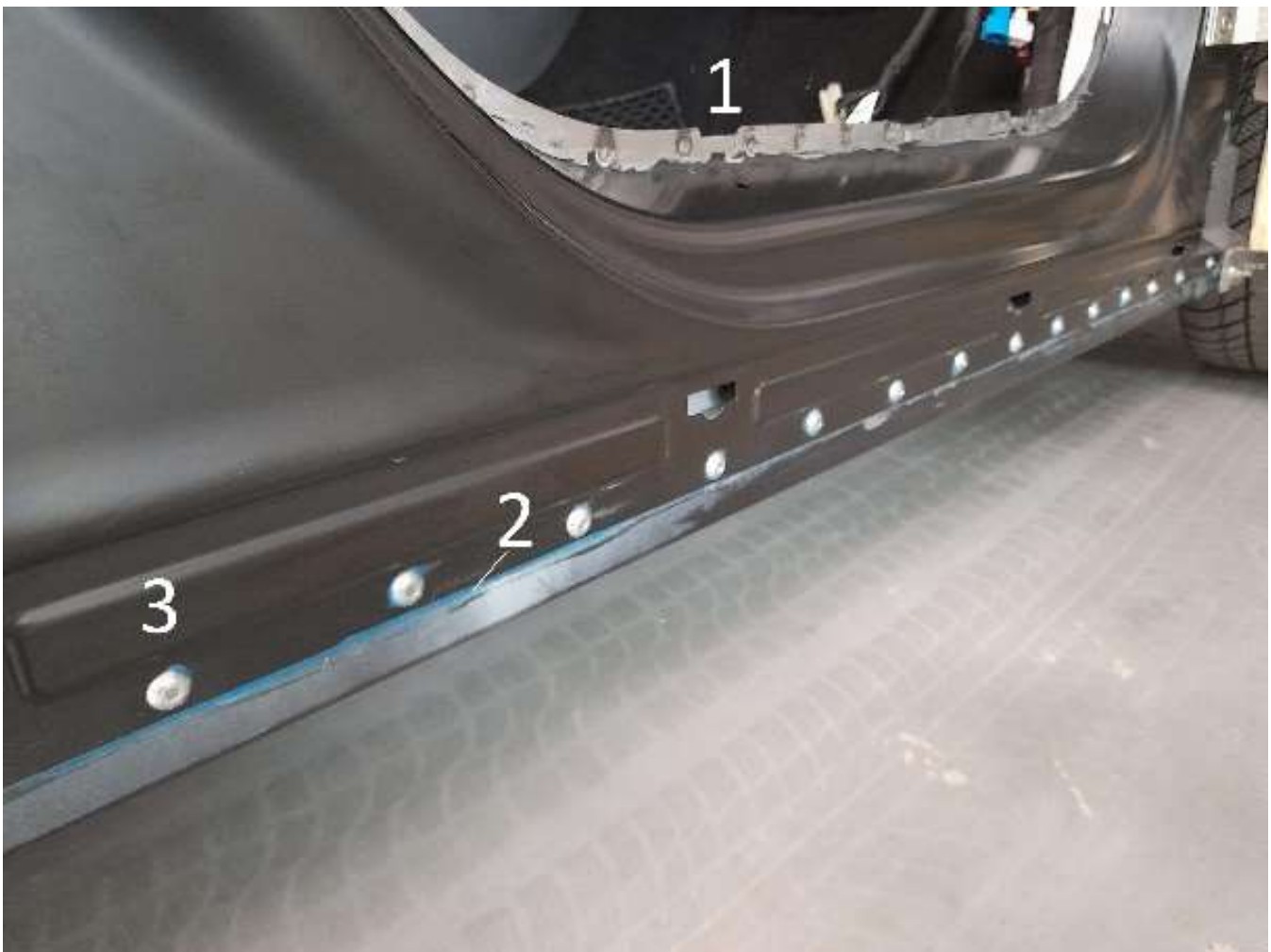

**Figure 2.** Joints used in car body repair; 1—weld; 2—adhesive; 3—classic rivet.

The aim of the research is to assess the quality of hybrid joints—sheet metal clinching and adhesive in various technological variants. The performed analyses will allow the formulation of technological guidelines related to this joining technology.

## 2. Materials and Methods

### 2.1. Samples

The plates used in this study were made of DC01 cold-rolled low-carbon steel. These sheets are used for the production of car body parts and household appliance parts. The chemical composition of the DC01 sheet, according to EN 102130, is shown in Table 1. The sheet has a tensile strength TS below 280 MPa.

**Table 1.** Chemical composition (%).

| Designation | Numerical Classification | C [%] | Mn [%]. | P [%] | S [%] | Si | Ti | Al | Nb |
|---|---|---|---|---|---|---|---|---|---|
| DC01 | 1.033 | ≤0.12 | ≤0.6 | ≤0.045 | ≤0.045 | - | - | - | - |

It was assumed that the samples used in the study should be as close as possible to the materials used in industrial conditions, so the thickness of the glued sheet was set at 0.7 mm. The adhesives available on the market are characterised by different bonding strengths, ranging from 2 MPa to 30 MPa. Due to the small thickness of the sheet and the relatively low tensile strength of DC01, it was decided to use strips of metal sheet with a width of 17 mm, which—assuming an area for bonding of 17 × 20 mm for different adhesives—gives bonding stresses in the sheet from 57 MPa for the weakest adhesives to 855 MPa for the strongest adhesives. In addition to the adhesive bonding, the sheet metal clinching will be utilised, the maximum stress of the specimens will be higher. This means that if a thin plate thickness is used, an adhesive with relatively low shear strength should be used. A hybrid adhesive was chosen, which has a lower strength than epoxy and methacrylate adhesives, but its setting time is shorter, and according to the manufacturer, the surface preparation does not significantly affect the quality of the bond. The bonding tests were performed using high-strength adhesive (Figure 3).

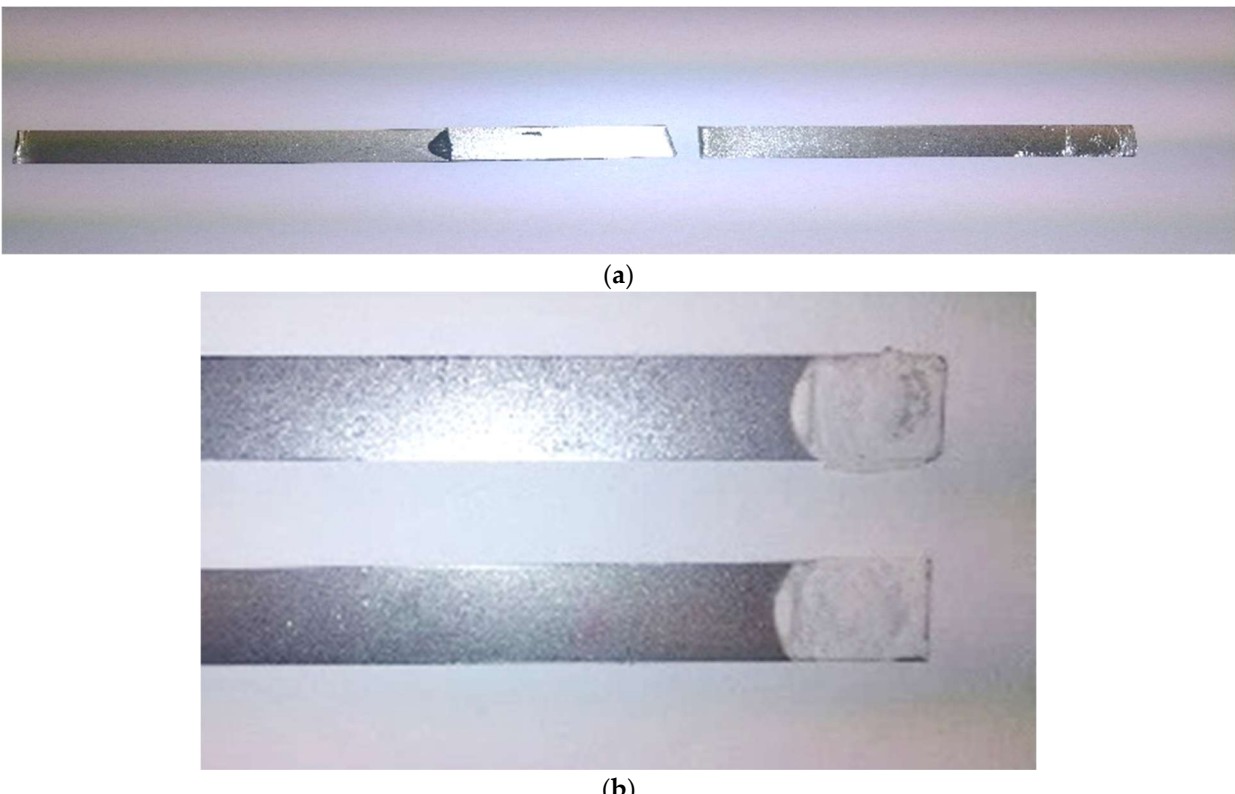

(**a**)

(**b**)

**Figure 3.** Comparison of ruptured specimens; (**a**) high-strength adhesive; (**b**) hybrid adhesive used in tests.

Before starting the tests, a series of specimens were prepared with different surface preparation. In the first case, the sheet was only degreased; in the second one, it was sandblasted with EK40 electrocorundum (P.P.H. REWA, KOLUSZKI, Poland) and degreased, while in the third case, the surfaces of the sample were treated with P80 sandpaper. The aim of this work was to select the surface preparation for bonding in further studies. The roughness parameters were examined and evaluated in all the samples. The arithmetic

mean deviation of the roughness profile ($R_a$) and the arithmetic mean of the absolute values of the height of the five highest peaks of the roughness profile and the depth of the five lowest roughness profile pits in the segmental segment ($R_z$) were investigated. All of the measurements were taken in accordance with the EN ISO 4287 and EN ISO 4288 standards. For the degreased surface, the roughness $R_a$ parameter was 0.49 μm, and $R_z$ was 2.6 μm; for the blasted surface, $R_a$ was 2.46 μm, and $R_z$ was 14.4 μm; while for the surface sanded with P80 grit paper, $R_a$ was 0.78 μm, and $R_z$ was 3.88 μm—Figure 4.

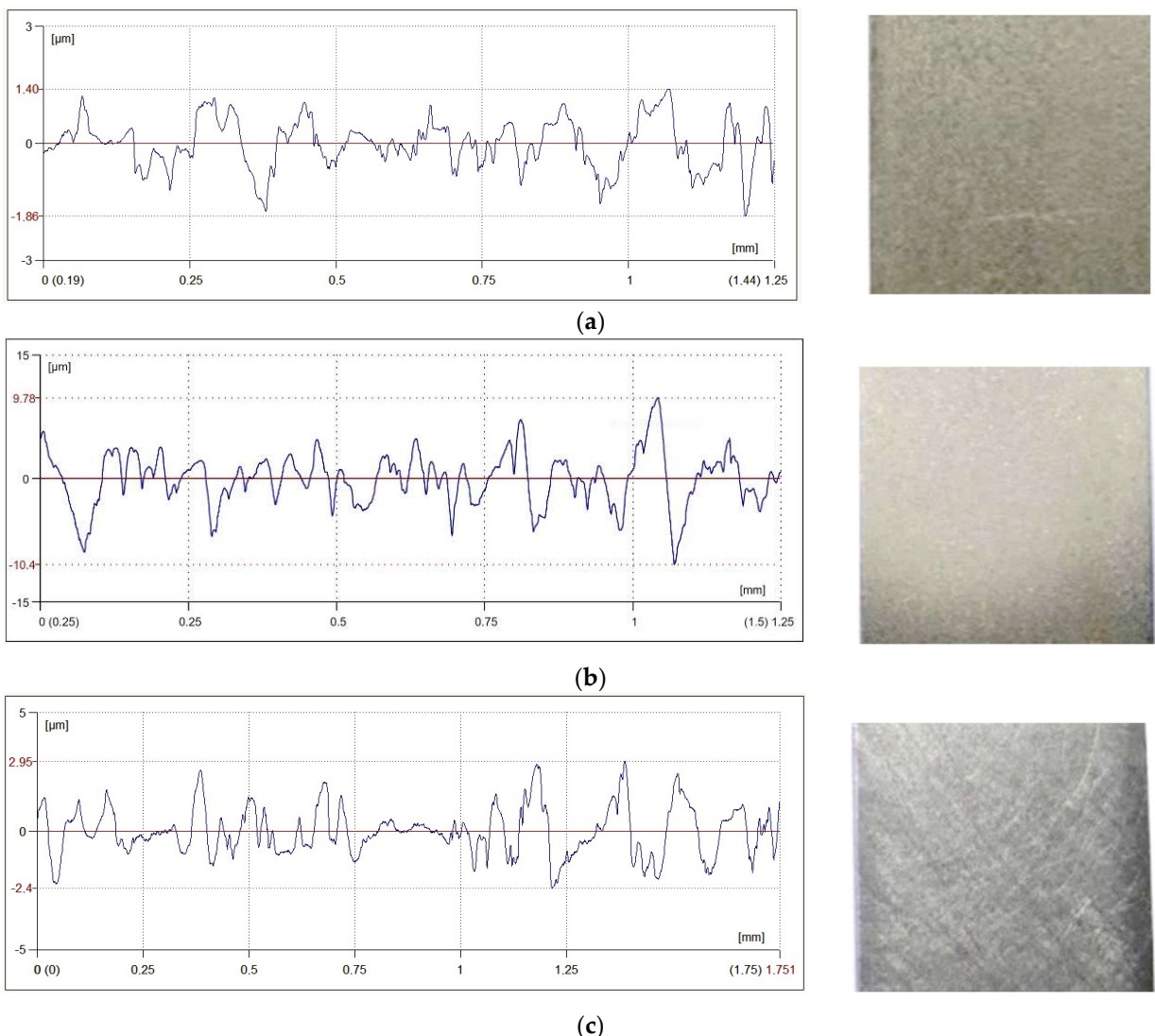

**Figure 4.** Roughness profile and view of the surface of the samples; (**a**) for the degreased surface, (**b**) for the abrasive blasted surface and (**c**) for the surface sanded with P80 paper.

### 2.2. Research Plan

In the studies, four main methods of sample preparation were distinguished. After the adhesive joints had been performed and fully formed, their quality was initially evaluated using the ultrasonic method. Following that, the joints were ruptured by registering the maximum force. The course of the individual research stages is shown in Figure 5.

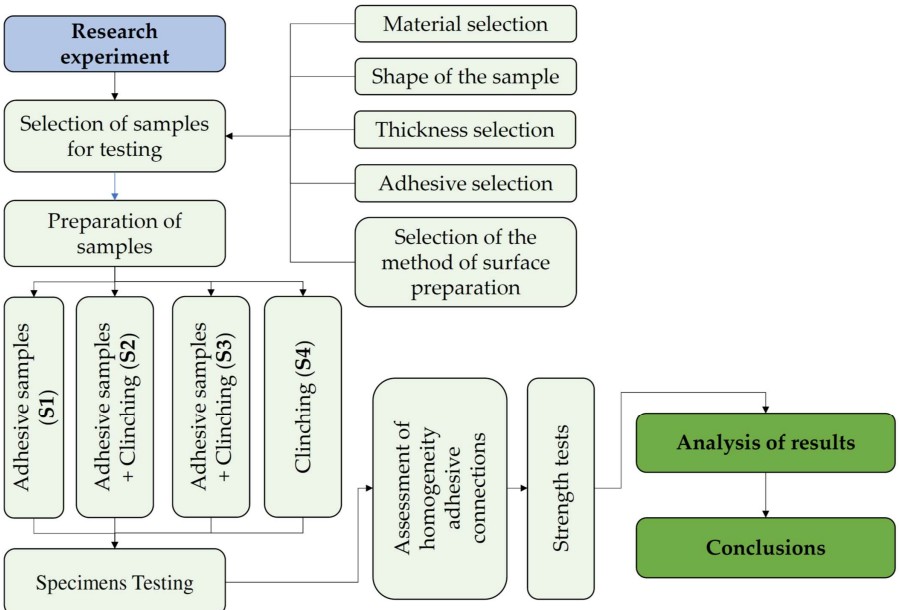

**Figure 5.** Plan of the research.

## 2.3. Sample Preparation

According to the plan (Figure 5), four series of 50 specimens were prepared. In the first test series, adhesive specimens were made, in which the surface was only degreased. The second test series consisted of specimens that, after the bonding and full curing of the joints, were additionally reinforced with sheet metal clinching. In the third test series, the clinching joints were made immediately after the adhesive was applied and the sheets were joined (Figure 6a). The last test series consisted of specimens made only with the use of a clinching joint, without adhesive (Figure 6b). During the joining process of the specimens, the splicing force values were measured as a function of the punch path. Example plots are shown in Figure 7. The list of test series is presented in Table 2. No differences were found for each of the tested measurement series. The displacement from 0 to 4 (mm) is the area of clearance compensation on the plate stamping station.

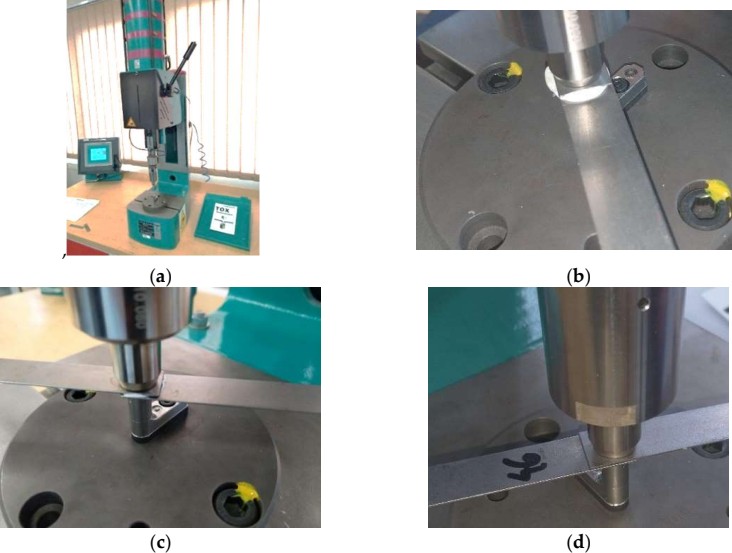

**Figure 6.** Execution of sheet metal clinching; (**a**) the device used to prepare clinching joint, (**b**) specimen from the first testing series, (**c**) specimen from the second testing series, (**d**) specimen from the third testing series.

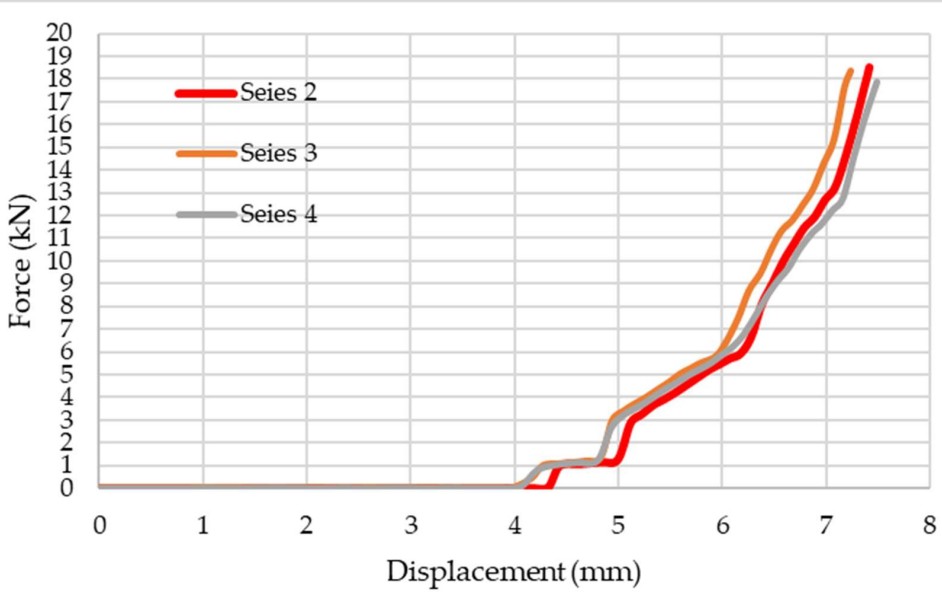

**Figure 7.** Plot of the dependence of the splicing force as a function of the punch path.

**Table 2.** Summary of research series.

| Designation of the Test Series | Type of Metal Sheet | Type of Adhesive | Clinching | Sample View |
|---|---|---|---|---|
| Series 1 | DC01 | Hybrid adhesive | Without Clinching | |
| Series 2 | DC01 | Hybrid Type of sheet metal | Clinching after the adhesive has cured | |
| Series 3 | DC01 | Hybrid adhesive | Clinching immediately after the application of the adhesive | |
| Series 4 | DC01 | Without adhesive | Only Clinching | |

### 2.4. Ultrasonic Tests

After the adhesive joints had been made, they were checked using the ultrasonic method to determine whether the shaped joints were of similar quality so as to indicate

possible poor-quality joints. The decibel drop between the first and the second pulse on the screen of the ultrasonic flaw detector, marked as R, used in the testing of the adhesive joints, was adopted as a measure of joint quality [1]. The ultrasonic wave velocity was determined for the tested material (DC01 steel) and amounted to 6325 m/s. Since the thickness of the sheet was 0.7 mm, an ultrasonic transducer with a frequency of 20 MHz was used in the tests. For this frequency, the wavelength was 0.32 mm, which was shorter than half of the sheet thickness. The parameters of the selected head are shown in Table 3.

**Table 3.** Parameters of the ultrasound transducer used.

| Parameter | | |
|---|---|---|
| frequency | 20 | MHz |
| diameter of the transducer | 3.15 | mm |
| effective diameter | 3.05 | mm |
| wave speed of tested material | 6325 | m/s |
| wavelength | 0.31 | mm |
| near field | 7.3 | mm |
| the coefficient of decreased decibels K | 0.87 | - |
| sin the angle of divergence of the beam | 0.09 | - |
| divergence angle degrees | 5.17 | O |
| distance from the transducer | 17 | mm |
| beam width | 3.1 | mm |

Since the near field was more than 7 mm, it was decided to use water delay. The test stand used is shown in Figure 8. The accuracy of the test stand was assessed by performing 30 measurements at the same point where the connection was made correctly and 30 measurements at the same point where the adhesive was not applied.

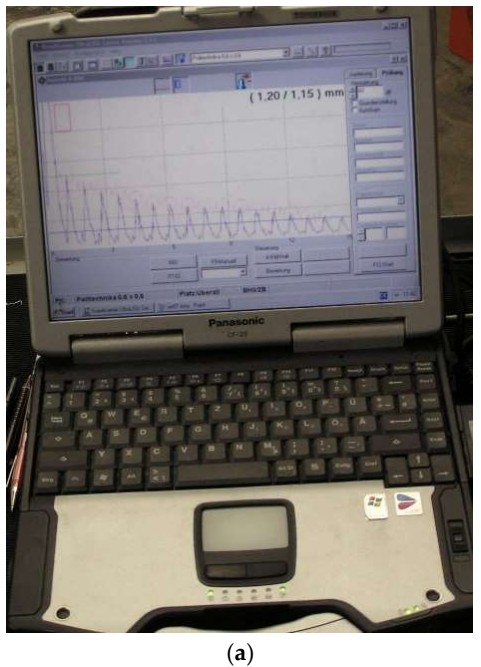

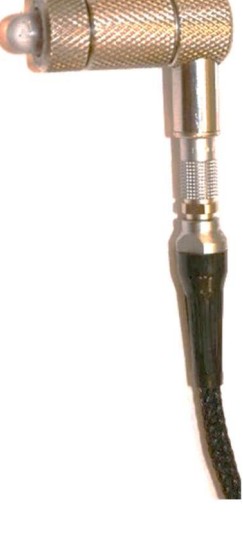

(**a**)          (**b**)

**Figure 8.** Ultrasonic equipment used in the research; (**a**) digital flaw detector, (**b**) ultrasonic head 20GX.

Figure 9 shows the ultrasound head and scheme of the view received on the flaw detector screen. The numbered echoes (4, 5, 6) are the multi-reflections occurring in the top sheet of the joint.

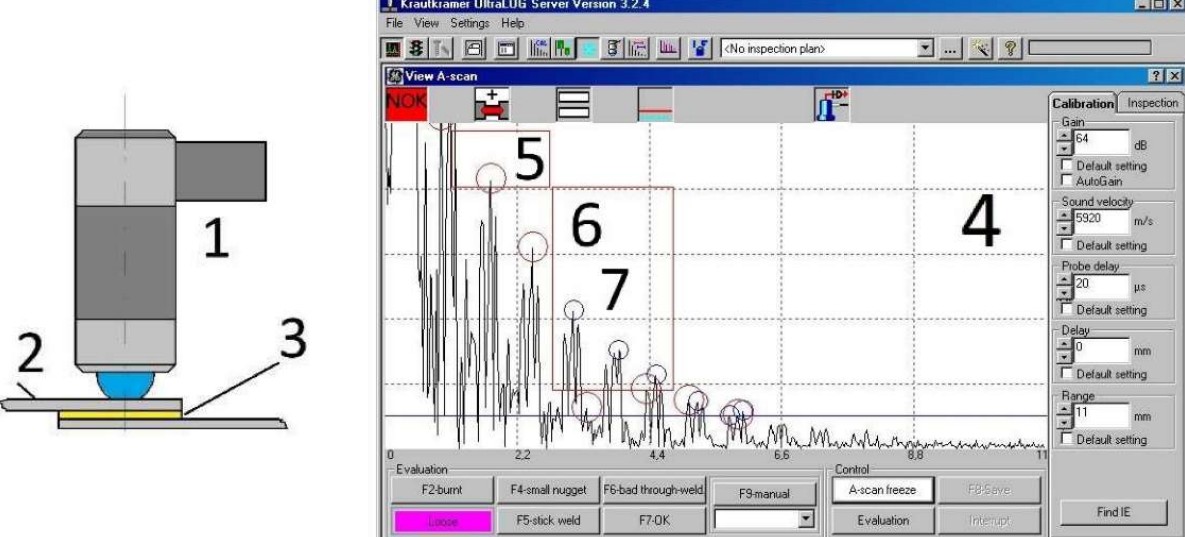

**Figure 9.** Ultrasonic testing; 1—ultrasonic probe; 2—stell DC01; 3—adhesive; 4—ultrasonic flaw detector screen; 5—first echo on the flaw detector screen (in this case $H_I$ = 80 %); 6—second echo on the flaw detector screen (in this case $H_{II}$ = 40 %); 7—third echo on the flaw detector screen (in this case $H_{III}$ = 9 %).

### 2.5. Strength Tests

The strength tests were carried out using A Cometech B1/E testing machine (Cometech Testing Machines Co., Taichung, Taiwan)—Figure 10. During the work, the maximum load of the tested joints was recorded.

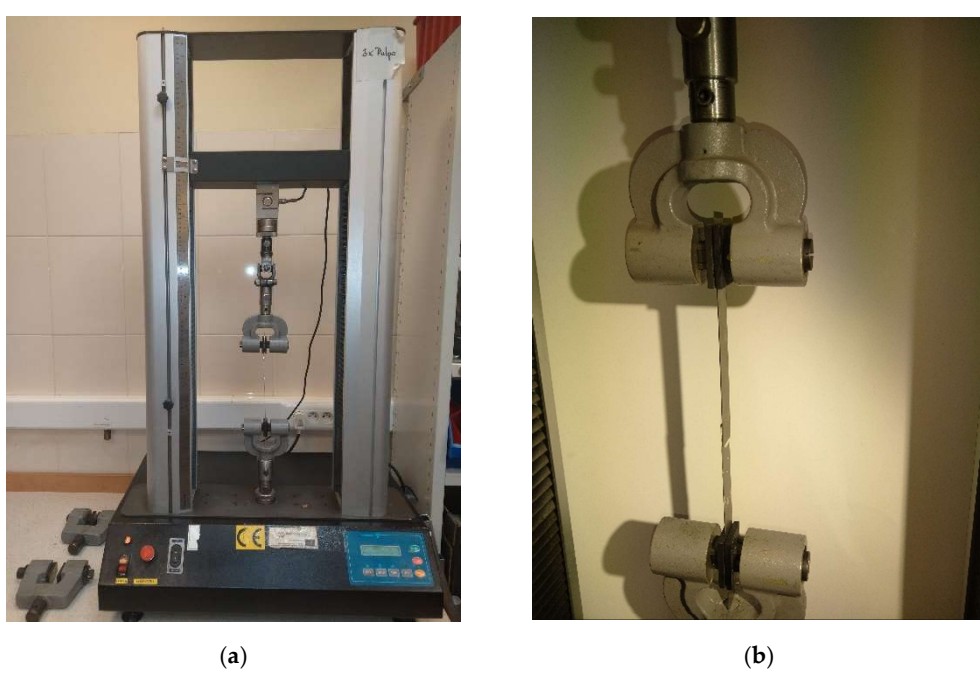

(**a**)          (**b**)

**Figure 10.** Testing machine; (**a**) general view, (**b**) specimen clamped.

The research was carried out for four series of measurements. The breaking force and the displacement of the clamp were measured in all series of measurements.

## 3. Results

### 3.1. Selection of Surface Preparation

As a preliminary study, the extent to which the surface preparation of the specimens affects the mechanical strength of the joint was evaluated. As part of this research, three series of specimens were made with different surface preparation for bonding. In the first one, the sheet was only degreased; in the second one, it was sandblasted with EK40 electrocorundum (P.P.H. REWA, KOLUSZKI, Poland) and degreased, while in the third case, the surfaces to be bonded were treated with P80 sandpaper and degreased. The results of the roughness measurements are shown in Figure 4. The results of the strength tests showed that all the joints were characterised by similar mechanical shear strength (Table 4). Taking into account the ways of joint preparation in the conditions of the automotive manufacturer and of the body repair shop, it was decided to choose the surface degreasing for further work.

**Table 4.** Influence of the preparation of the surface to be bonded the bonding strength of the joint.

|  | Sandblasting (N) | Sanding (N) | Degreasing (N) |
| --- | --- | --- | --- |
| Min | 432 | 432 | 438 |
| Max | 519 | 515 | 521 |
| Average | 477.05 | 480.25 | 486.4 |
| Deviation | 30.11 | 26.28 | 21.39 |
| Confidence interval $L_{0.1}$ | 52.06 | 45.44 | 36.99 |

The results of the tests are summarised in Figure 11. It was observed that cohesive rupture occurred in all the tested joints.

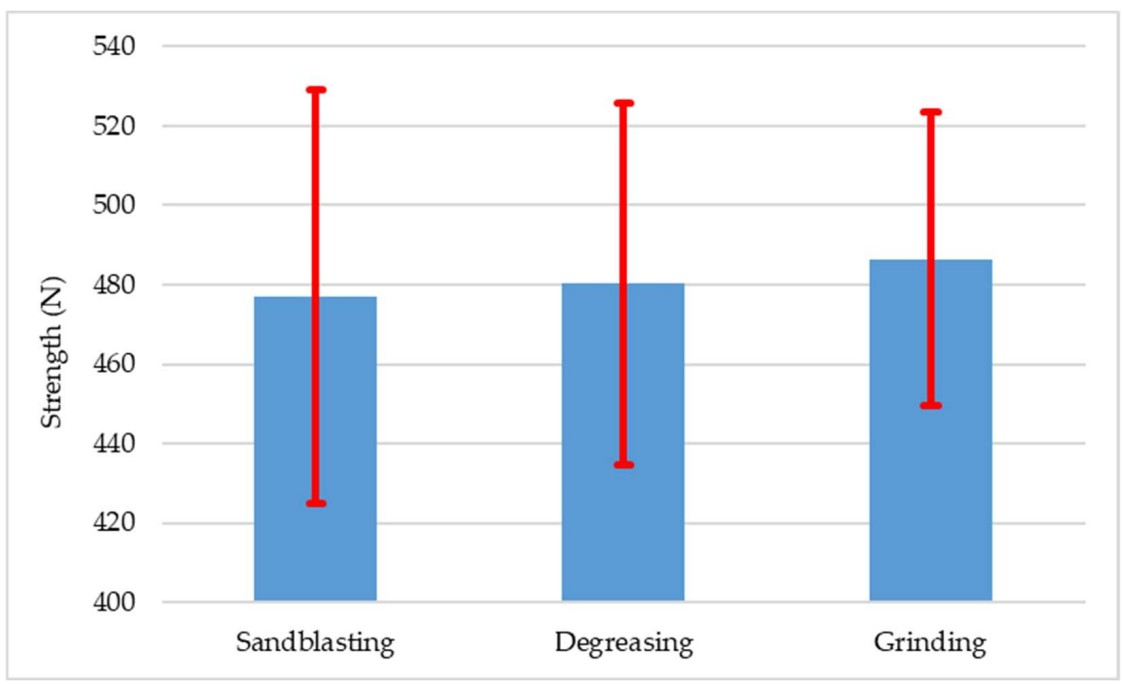

**Figure 11.** Results of preliminary studies.

### 3.2. Results of Ultrasonic Tests

The aim of the conducted ultrasonic tests was to assess the quality dispersion of the shaped joints. The properties of the materials tested—especially the small thickness sheet metal—made the testing extremely difficult [1]. In the first step, the measurement errors were evaluated. The errors were determined for the first five pulses from the joint area and

the sheet without adhesive applied. The results are shown in Tables 5 and 6. During these measurements, the height of the first pulse amounted to 80% of the height of the ultrasonic flaw detector screen was set.

**Table 5.** Pulse heights on the flaw detector screen—determination of measurement errors in the defect area (the complete set of results are available in the Appendix A, Table A1).

|  | $H_{II}$ (%) | $H_{III}$ (%) | $H_{IV}$ (%) | $H_V$ (%) |
|---|---|---|---|---|
| Average | 69.8 | 58.8 | 50.0 | 41.6 |
| Deviation | 1.03 | 2.08 | 0.98 | 1.97 |
| T-student coefficient 10% | 1.66 | 1.67 | 1.67 | 1.67 |
| Confidence interval $L_{0,1}$ | 1.72 | 3.47 | 1.64 | 3.29 |

**Table 6.** Pulse heights on the flaw detector screen—determination of measurement errors in the defect area (the complete set of results are available in the Appendix A, Table A2).

|  | $H_{II}$ (%) | $H_{III}$ (%) | $H_{IV}$ (%) | $H_V$ (%) |
|---|---|---|---|---|
| Average | 63.4 | 49.9 | 38.7 | 28.6 |
| Deviation | 5.27 | 4.30 | 3.31 | 3.67 |
| T-student coefficient 10% | 1.67 | 1.67 | 1.67 | 1.67 |
| Confidence interval $L_{0,1}$ | 8.78 | 7.18 | 5.53 | 6.11 |

The presented results of ultrasonic testing confirmed that the measurement errors obtained during ultrasonic testing are small and will not significantly affect the results of further work in both the area of joints and areas where the adhesive was not applied.

Ultrasonic testing was performed for the first three test groups of the sample because bonding was used in these tests. The results of the ultrasonic tests are shown in Table 7.

**Table 7.** Ultrasonic measure of joint quality R for the first three measurement runs (the complete set of results are available in the Appendix A, Table A3).

|  | Series 1 | Series 2 | Series 3 |
|---|---|---|---|
| Min | 1.784 | 1.843 | 1.802 |
| Max | 3.307 | 3.383 | 3.212 |
| Average | 2.572 | 2.654 | 2.513 |
| Standard deviation | 0.432 | 0.466 | 0.389 |

The results of the ultrasonic tests show that the quality of the adhesive joints did not differ significantly. This means that all the joints were made correctly, and the adhesive bonding conditions were constant.

*3.3. Results of Bonding Strength*

Strength tests were conducted for all the prepared specimens. Figure 12 shows examples of force–displacement plots for specimens from each measurement series, while the results are summarised in Table 8. The view of exemplary samples after breaking is shown in Figure 9, while the loaded samples during the strength tests are shown in Figure 13.

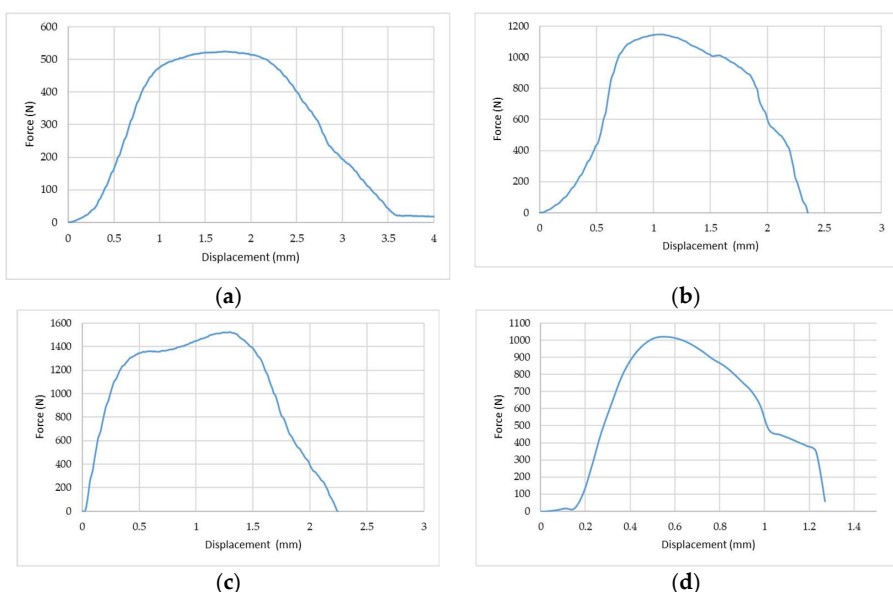

**Figure 12.** Example force vs. distance plots for (**a**) first test series, (**b**) second test series, (**c**) third test series, and (**d**) fourth test series.

**Table 8.** Shear force obtained during strength tests (the complete set of results are available in the Appendix A, Table A4).

|  | Series 1 (N) | Series 2 (N) | Series 3 (N) | Series 4 (N) |
|---|---|---|---|---|
| Min | 476 | 1085 | 1486 | 965 |
| Max | 622 | 1247 | 1749 | 1082 |
| Average | 551.64 | 1166.86 | 1617.1 | 1013.08 |
| Standard deviation | 44.41 | 44.00 | 79.09 | 34.91 |

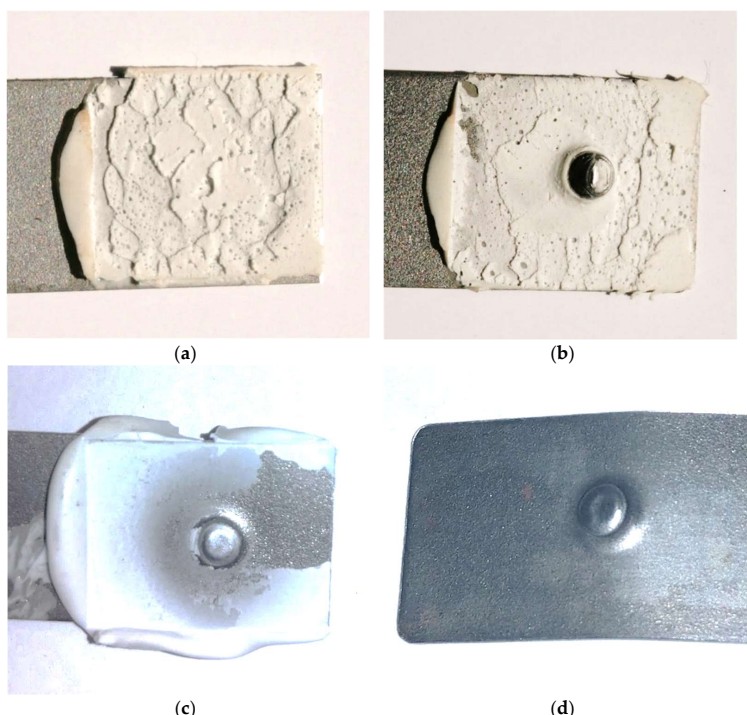

**Figure 13.** Example view of samples after shear test for: (**a**) first test series, (**b**) second test series, (**c**) third test series, and (**d**) fourth test series.



The force vs. displacement diagrams are typical for an elastic adhesive. Stresses gradually build up after reaching the maximum value and there is a gradual decrease after breaking the connection. For a thicker layer of glue (third test series), it can be seen that the high stress value is maintained for about 2 mm displacement. The nature of the diagram is typical for hybrid and flexible adhesives [29,30] with higher elongation; for structural adhesives, the nature of the diagram is different [31,32]. Structural adhesives decrease more rapidly after reaching the maximum value, similar to clinching.

The thickness of the adhesive after tearing was measured during the tests. The thickness measurements were performed using the Karl Deutsch KD2050 leptoscope (Karl Deutsch, Wuppertal, Germany). The average thickness of the adhesive in the first and second test series was 161 micron meters on one surface. In the third test series, however, the average adhesive thickness was 20.3 microns. The distribution of the adhesive thickness on the bonded surface of the first and third test series is shown in Figure 14.

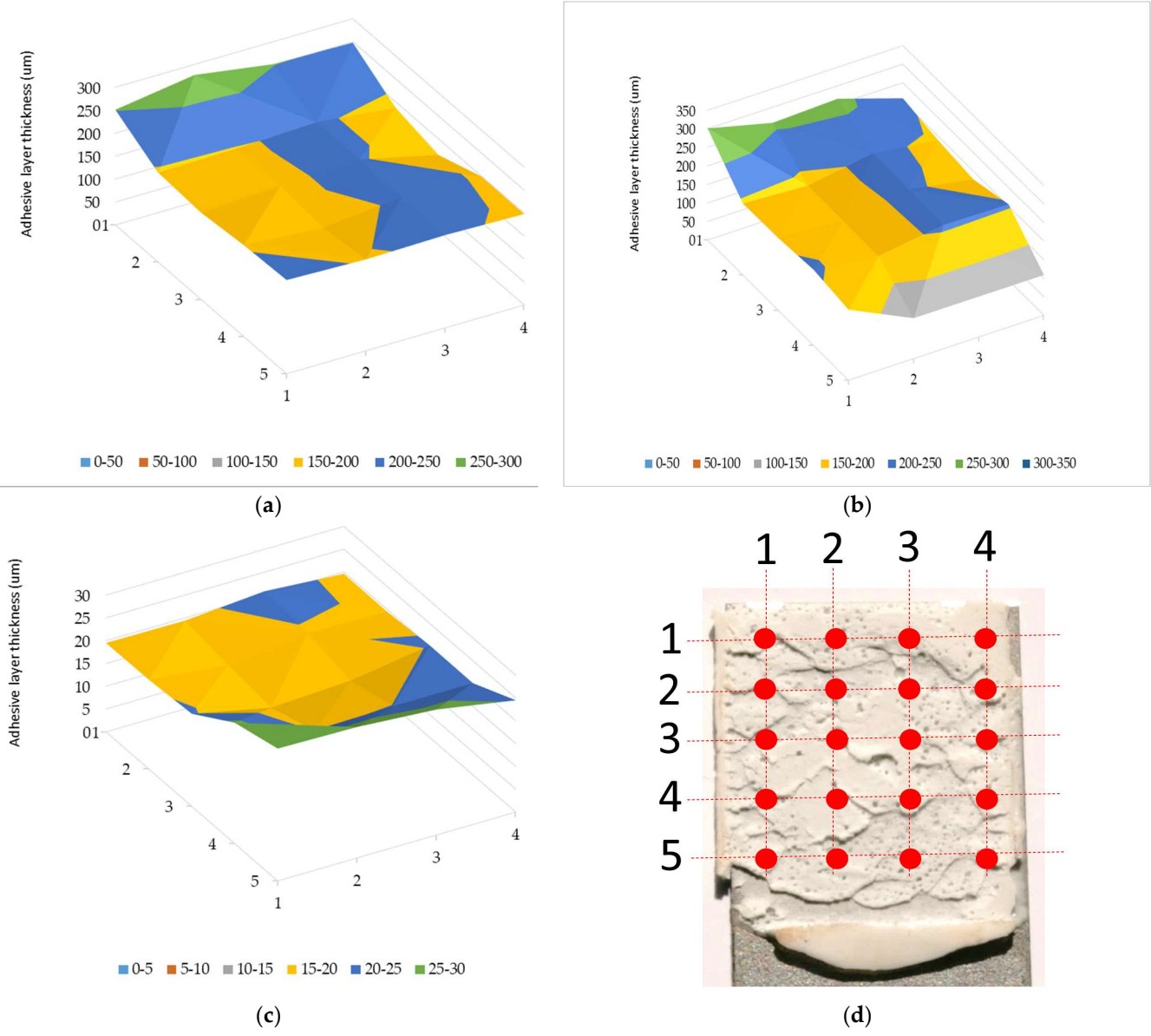

**Figure 14.** Adhesive thickness after joint shear test for: (**a**) first test series, (**b**) second series, (**c**) third test series, (**d**) the origin of the measurement points.

The research has confirmed that the usage of sheet metal clinching on an adhesive joint immediately after production reduces the thickness of the adhesive joint. Reducing the thickness of the adhesive joint resulted in an increase in the strength of the joint, which is in line with other works [33]. In a properly designed and constructed adhesive joint, cohesive rupture is preferred when the allowable stresses are exceeded. Resistance to cohesive rupture should be higher than cohesive rupture. An increase in the thickness of the adhesive bond, especially in polyurethane, MS Polymer, Hybrid adhesives, causes the area susceptible to rupture to increase, while at the same time, the nature of the stresses in the bond area causes the strength of the bond to decrease.

## 4. Conclusions

The conducted tests confirmed that it is reasonable to perform hybrid joints for steel sheets and hybrid adhesives. The average shear force of adhesive joints was 476 N, while the average force for a sheet metal clinching joint was 965 N. The difference between these forces is almost 500 N. Significantly higher strength than the one found in the adhesive and clinching joints was obtained for the hybrid joint. For the specimens in which the clinching joint was developed immediately after the adhesive joint, the strength was 312% higher than for the adhesive joint and 154% higher for the clinching joint. The study showed that the timing of the clinching joint is important. Preferably, the joint should be developed immediately after the adhesive bond. This increases the mechanical strength by 37%. If the splices are made immediately after the adhesive bonds, the adhesive bond is much thinner. For the hybrid joints in which the clinching was performed after the adhesive joints were fully formed, the thickness of the adhesive was, on average, 322 um, while in those specimens in which the sheet metal clinching was performed right after the adhesive joints were formed, it is about eight times less and amounts to 40.6 um. The thickness of the adhesive affects the quality of the connection. Greater thickness reduces the quality of the connection. The proposed technology not only increases the strength of the bond but can also lead to a reduction in adhesive consumption. This is not only an economic aspect but also an ecological one. The ecological aspect results from both reduced adhesive consumption and the decreased weight of the finished product, which directly translates to the emission of harmful substances [34,35]. In further research, it is planned to combine materials with different acoustic properties. Adhesives with low damping coefficients and aluminium sheets will be used.

**Author Contributions:** Conceptualisation, J.K.; methodology, J.K., W.M. and W.S.; software, K.S.; validation, J.K., W.M. and K.S.; formal analysis, J.K. and W.S.; investigation, J.K.; resources, J.K. and W.M.; data curation, J.K. and D.W.; writing—original draft preparation, J.K. and W.S.; writing—review and editing, J.K.; visualization, J.K. and D.W.; supervision, M.N.; project administration, K.S.; funding acquisition, K.S. and M.N. All authors have read and agreed to the published version of the manuscript.

**Funding:** This work was supported by the Polish National Centre for Research and Development under the grant—decision no. DWP/TECHMATSTRATEG-III/136/2020.

**Institutional Review Board Statement:** Not applicable.

**Informed Consent Statement:** Not applicable.

**Data Availability Statement:** The data presented in this study are available upon request from the corresponding author.

**Conflicts of Interest:** The authors declare no conflict of interest.

## Appendix A

**Table A1.** Impulse heights on the ultrasonic flaw detector screen—determination of measurement errors in the unglued area.

| $H_I$ (%) | $H_{II}$ (%) | $H_{III}$ (%) | $H_{IV}$ (%) | $H_V$ (%) |
|---|---|---|---|---|
| 80 | 70 | 62 | 51 | 43 |
| 80 | 71 | 61 | 49 | 39 |
| 80 | 72 | 55 | 51 | 44 |
| 80 | 70 | 57 | 49 | 39 |
| 80 | 72 | 62 | 51 | 44 |
| 80 | 68 | 58 | 50 | 42 |
| 80 | 69 | 61 | 49 | 40 |
| 80 | 69 | 58 | 51 | 39 |
| 80 | 70 | 57 | 50 | 42 |
| 80 | 69 | 60 | 49 | 39 |
| 80 | 71 | 59 | 51 | 40 |
| 80 | 69 | 60 | 50 | 43 |
| 80 | 70 | 60 | 50 | 44 |
| 80 | 70 | 57 | 50 | 40 |
| 80 | 70 | 60 | 49 | 44 |
| 80 | 69 | 60 | 54 | 38 |
| 80 | 68 | 60 | 49 | 40 |
| 80 | 68 | 58 | 49 | 42 |
| 80 | 69 | 60 | 50 | 39 |
| 80 | 70 | 59 | 49 | 42 |
| 80 | 69 | 60 | 51 | 43 |
| 80 | 69 | 55 | 49 | 43 |
| 80 | 71 | 57 | 50 | 43 |
| 80 | 69 | 55 | 50 | 44 |
| 80 | 68 | 56 | 50 | 43 |
| 80 | 70 | 60 | 49 | 43 |
| 80 | 70 | 60 | 50 | 44 |
| 80 | 71 | 60 | 50 | 43 |
| 80 | 70 | 62 | 50 | 43 |
| 80 | 69 | 57 | 50 | 40 |

**Table A2.** Impulse heights on the ultrasonic flaw detector screen—determination of measurement errors in the glue joint area.

| $H_I$ (%) | $H_{II}$ (%) | $H_{III}$ (%) | $H_{IV}$ (%) | $H_V$ (%) |
|---|---|---|---|---|
| 80 | 64 | 50 | 39 | 30 |
| 80 | 70 | 54 | 42 | 33 |
| 80 | 63 | 47 | 37 | 20 |
| 80 | 58 | 46 | 34 | 27 |
| 80 | 65 | 55 | 43 | 31 |
| 80 | 59 | 49 | 39 | 28 |
| 80 | 65 | 54 | 43 | 31 |
| 80 | 65 | 54 | 43 | 31 |
| 80 | 70 | 56 | 42 | 33 |
| 80 | 59 | 49 | 39 | 28 |
| 80 | 69 | 54 | 41 | 31 |
| 80 | 70 | 54 | 41 | 33 |
| 80 | 63 | 48 | 37 | 28 |
| 80 | 65 | 54 | 43 | 31 |
| 80 | 69 | 54 | 41 | 31 |
| 80 | 63 | 49 | 37 | 28 |
| 80 | 57 | 43 | 33 | 25 |

**Table A2.** *Cont.*

| H$_I$ (%) | H$_{II}$ (%) | H$_{III}$ (%) | H$_{IV}$ (%) | H$_V$ (%) |
|---|---|---|---|---|
| 80 | 71 | 56 | 43 | 33 |
| 80 | 58 | 45 | 35 | 27 |
| 80 | 57 | 42 | 33 | 18 |
| 80 | 57 | 43 | 33 | 25 |
| 80 | 58 | 45 | 34 | 26 |
| 80 | 64 | 49 | 38 | 30 |
| 80 | 70 | 53 | 40 | 31 |
| 80 | 70 | 52 | 41 | 23 |
| 80 | 64 | 51 | 38 | 30 |
| 80 | 56 | 47 | 37 | 30 |
| 80 | 55 | 46 | 36 | 25 |
| 80 | 70 | 55 | 43 | 33 |
| 80 | 56 | 43 | 34 | 25 |

**Table A3.** Ultrasonic R joint quality measure for the first three measurement series.

|  | Series 1 | Series 2 | Series 3 |
|---|---|---|---|
| 1 | 1.788 | 2.095 | 2.300 |
| 2 | 2.440 | 1.882 | 2.471 |
| 3 | 3.098 | 2.708 | 2.408 |
| 4 | 2.552 | 3.362 | 2.534 |
| 5 | 2.642 | 2.923 | 2.368 |
| 6 | 2.499 | 3.206 | 2.636 |
| 7 | 3.161 | 2.781 | 2.796 |
| 8 | 3.121 | 3.152 | 2.432 |
| 9 | 1.784 | 3.284 | 2.532 |
| 10 | 2.741 | 2.392 | 2.810 |
| 11 | 1.938 | 2.448 | 2.667 |
| 12 | 2.607 | 2.890 | 2.876 |
| 13 | 2.167 | 2.923 | 3.198 |
| 14 | 3.178 | 1.998 | 2.588 |
| 15 | 2.392 | 1.843 | 2.525 |
| 16 | 2.602 | 2.144 | 2.255 |
| 17 | 2.956 | 3.335 | 2.792 |
| 18 | 3.171 | 3.368 | 2.172 |
| 19 | 2.732 | 3.383 | 2.283 |
| 20 | 2.246 | 3.152 | 1.872 |
| 21 | 2.952 | 2.653 | 2.486 |
| 22 | 2.899 | 2.330 | 1.950 |
| 23 | 2.499 | 2.201 | 3.070 |
| 24 | 3.152 | 2.868 | 2.032 |
| 25 | 2.362 | 2.199 | 2.201 |
| 26 | 3.307 | 2.095 | 3.207 |
| 27 | 1.938 | 3.118 | 2.010 |
| 28 | 2.612 | 2.322 | 2.969 |
| 29 | 2.694 | 2.289 | 1.994 |
| 30 | 3.013 | 2.671 | 1.802 |
| 31 | 3.098 | 2.082 | 3.016 |
| 32 | 2.036 | 3.115 | 2.523 |
| 33 | 1.994 | 2.384 | 1.955 |
| 34 | 3.208 | 2.197 | 2.468 |
| 35 | 2.724 | 2.138 | 2.242 |
| 36 | 2.416 | 3.313 | 3.212 |

**Table A3.** *Cont.*

|  | **Series 1** | **Series 2** | **Series 3** |
|---|---|---|---|
| 37 | 2.713 | 3.098 | 2.274 |
| 38 | 1.911 | 3.167 | 2.193 |
| 39 | 2.838 | 3.343 | 2.960 |
| 40 | 2.138 | 2.499 | 3.015 |
| 41 | 2.753 | 2.392 | 2.792 |
| 42 | 2.751 | 2.901 | 1.948 |
| 43 | 1.855 | 2.308 | 2.414 |
| 44 | 3.174 | 2.267 | 1.899 |
| 45 | 2.167 | 3.121 | 2.249 |
| 46 | 2.267 | 2.342 | 2.975 |
| 47 | 2.684 | 3.076 | 2.742 |
| 48 | 2.107 | 2.227 | 3.032 |
| 49 | 2.121 | 2.265 | 2.885 |
| 50 | 2.425 | 2.461 | 2.599 |

**Table A4.** Strength test results—shear force value (N).

|  | **Series 1** | **Series 2** | **Series 3** | **Series 4** |
|---|---|---|---|---|
| 1 | 514 | 1238 | 1624 | 1018 |
| 2 | 564 | 1161 | 1551 | 1032 |
| 3 | 622 | 1231 | 1486 | 974 |
| 4 | 560 | 1110 | 1652 | 1042 |
| 5 | 621 | 1160 | 1529 | 987 |
| 6 | 486 | 1123 | 1544 | 1063 |
| 7 | 500 | 1189 | 1528 | 1027 |
| 8 | 476 | 1094 | 1678 | 984 |
| 9 | 516 | 1100 | 1638 | 1008 |
| 10 | 600 | 1174 | 1706 | 1058 |
| 11 | 494 | 1145 | 1488 | 1082 |
| 12 | 596 | 1243 | 1629 | 969 |
| 13 | 554 | 1198 | 1562 | 1044 |
| 14 | 595 | 1215 | 1574 | 1037 |
| 15 | 478 | 1160 | 1545 | 1062 |
| 16 | 553 | 1154 | 1707 | 1005 |
| 17 | 586 | 1099 | 1707 | 985 |
| 18 | 585 | 1176 | 1673 | 1018 |
| 19 | 547 | 1202 | 1713 | 973 |
| 20 | 517 | 1195 | 1695 | 993 |
| 21 | 583 | 1152 | 1564 | 1028 |
| 22 | 568 | 1146 | 1720 | 1002 |
| 23 | 585 | 1097 | 1487 | 1024 |
| 24 | 523 | 1134 | 1652 | 1062 |
| 25 | 488 | 1127 | 1536 | 965 |
| 26 | 495 | 1233 | 1572 | 979 |
| 27 | 566 | 1236 | 1558 | 987 |
| 28 | 588 | 1200 | 1723 | 992 |
| 29 | 526 | 1247 | 1569 | 1014 |
| 30 | 602 | 1175 | 1517 | 1074 |
| 31 | 549 | 1185 | 1503 | 1044 |
| 32 | 604 | 1180 | 1673 | 972 |
| 33 | 486 | 1155 | 1642 | 970 |
| 34 | 573 | 1203 | 1660 | 1068 |
| 35 | 495 | 1107 | 1535 | 993 |
| 36 | 490 | 1135 | 1520 | 968 |
| 37 | 610 | 1142 | 1731 | 977 |

**Table A4.** *Cont.*

|  | Series 1 | Series 2 | Series 3 | Series 4 |
|---|---|---|---|---|
| 38 | 620 | 1181 | 1722 | 1059 |
| 39 | 488 | 1202 | 1517 | 977 |
| 40 | 595 | 1120 | 1632 | 1007 |
| 41 | 584 | 1164 | 1554 | 980 |
| 42 | 592 | 1237 | 1694 | 1082 |
| 43 | 507 | 1120 | 1687 | 1023 |
| 44 | 494 | 1208 | 1710 | 1040 |
| 45 | 591 | 1216 | 1749 | 981 |
| 46 | 586 | 1137 | 1610 | 994 |
| 47 | 554 | 1139 | 1634 | 984 |
| 48 | 547 | 1152 | 1651 | 990 |
| 49 | 562 | 1161 | 1737 | 1065 |
| 50 | 567 | 1085 | 1567 | 992 |

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
