# Peer review of "Quality Tests of Hybrid Joint–Clinching and Adhesive—Case Study"

_applsci, doi:10.3390/app122211782_

Round 1

Reviewer 1 Report

In this paper, the authors investigate a hybrid metal joint technique by combining clinching and adhesive and evaluate the joint strength performance. They tested four types of joints: adhesive alone, clinching alone, clinching immediately after adhesive, and clinching after adhesive cure. They conclude that the strength of the hybrid joint immediately after adhesive is on average 2.8 times higher than the adhesive joint alone and 1.6 times higher than the clinch joint alone. The strength of this type of hybrid joint is also increased by 50% compared to the hybrid joint after the adhesive cures.

Overall, the research strategy makes sense and the results are obvious. However the explanation and interpretation are not convincing enough. I recommend a minor revision of the manuscript and suggest that the authors revise the following points

1.        In introduction:

The authors introduced the different joining techniques including adhesive bonding, welding, clinching and riveting. A few references were cited, especially for adhesive bonding and clinching. However, there no analysis and comments on the works of the references. The content is not well organized.

The paragraph from line 82 to line 95 is a complete copy of the previous text from line 54 to line 67. It should be canceled.

2.        For the ultrasonic test, the test method is not clearly described and the aim of the test is not well explained:

What exactly does it mean (line 188) “The accuracy of the test stand was evaluated by taking 30 measurements on the test specimen where the joint was made correctly and the adhesive was not applied”? Is this a calibration of the test stand? Were the measurements made at 30 different locations on the joint or 30 times at the same location? 

If the coupling between the samples and the ultrasonic probe was realized using a head with a water chamber, is it the reason of the measurement errors?

In Table 3, what means “the coefficient of decrease of decibels K”?

It would be better to move Figure 11 from Section 3.2 to Section 2.4 as the testing method description. It is necessary to give an ultrasonic data signal example with its coordinates and units instead of a sketch indication. Please clarify if the numbered echoes (4 ,5, 6) are the multi-reflections occurring in the top sheet of the joint.

What means (line 223) "especially small thicknesses - making testing very difficult"? Does it refer to sheet thickness or adhesive thickness?

What means the defect area (in Tables 5 and 6)?  What represents the joint quality R (in Table 7) and how it was defined?

What was exactly determined from the ultrasonic echography measurements (the disbond or the adhesive quality)? If the results (Tables 5,A1, 6, A2) can confirm the detecting between the unglued and glued areas, there is no evidence that such test can determine the adhesive quality (affecting the final strength) which depends on not only the presence of the adhesive but also the adhesive properties such as its cohesion, density, cure process…

3.        In Section 3.3:

The abscissa unit (position ?) is missing from Figure 14 (a), (b) and (c).

Was the average thickness of the adhesive after tearing determined from the surface roughness profile?

Line 283, “Reducing the thickness of the adhesive joint resulted in an increase in the strength of the joint” - - Can the authors provide any explanation for this confirmation?

4.        In conclusion:

The author erroneously used the minimum shear force data (in Table 8) for the average shear force.

The authors concluded (line 299) again that the thickness of the adhesive affects the joint quality. Can the authors explain why?

The authors stated that reducing the consumption of adhesives has an economic and ecological benefit. The question is does the technique will really reduce the adhesive consumption?  Can the adhesive spilled by clinching be recovered and reused?

Line 305 - In future research, what is the relevance of using materials with different acoustic properties to the quality of the hybrid joint? What does a low damping adhesive mean?

Author Response

Dear Reviewer,
Thank you for reviewing the publication,
Please find attached in the pdf file the responses to the comments made.
The changes have also been included in the text and marked with a yellow background.
Yours sincerely

Reviewer 2 Report

General Comments

Overall, this is an excellent paper. The study was done carefully, and the conclusions are clear. I have no technical comments or suggestions.

The English is only fair. Given that I have no comments about the science, I included a bunch of comments (mostly minor) about the English below.

Specific Comments

The shear strengths given in the abstract are the minimum shear strengths in Table 8. Is this what was meant? Same issue in line 287: minimum and average are mixed up.

Line 37: “Bonding allow to join” should be “Bonding allows us to join” or “Bonding allows the joining of”

Line 40: “To overcome above disadvantages”. Here and in many other places, articles are incorrectly used. “To overcome the above disadvantages,”.

Line 43: “is finding frequently its application” is very awkward. Perhaps “in frequently applied”.

Line 50: “The works on the improvement of” would be better as “Improvements of” or even just “Improved”. Perhaps best would be “Joining technologies in motor vehicles are being improved all the time”.

I would move lines 68–71 to earlier, when you first introduce clinching (around line 51).

Line 77: “30÷60%”. Use a dash, and not a divide symbol.

Line 100: “the possibility of making connections only in non-visible areas”. I don’t quite understand the point.

Line 105: “allow to formulate” should be “allow us to formulate” or “allow the formulation of”

Line 140: Is “skimmed” the same as “degreased”? Unclear.

Line 161: Is “process force” in line 161 the same as “splicing force” in the caption of Fig. 7?

Line 174: “After the adhesive joints had been made, it was checked” should be “After the adhesive joints had been made, they were checked”

Line 179: “Since the thickness of the sheet equal 0.7 mm” should be “Since the thickness of the sheet equaled 0.7 mm” or better “Since the thickness of the sheet was 0.7 mm”.

Table 3: Giving some of these parameters to five or six significant figures is ridiculous. I doubt you know their values that accurately.

Line 202: What is a “gland”? This word does not appear in the paper except in Line 202.

Lines 205 to 210: Some places in the paper, including in these lines and in the introduction, you seem to repeat things you have said before.

Table 4: Does “grinding” mean the same as “sanding”?

Line 219: If you need to shorten the paper, Figure 10 and Table 4 provide the same information. One could be eliminated.

Table 8: Again, I doubt that it is appropriate to give six significant figures for these values.

Line 274: “micron meter”. Do you mean “micrometer” or “micron”?

I didn’t get much from the appendix. I suggest deleting it.

Author Response

Dear Reviewer,

Thank you for reviewing the publication,

The changes have been included in the text and marked with a yellow background.

Yours sincerely

Jakub Kowalczyk